# How does social capital affect individual health among the elderly in rural China?— Mediating effect analysis of physical exercise and positive attitude

**Hang Liang[1], Zhang Yue** [1]* **, Erpeng Liu[2], Nan Xiang[1]**

**1** School of Public Administration, Zhongnan University of Economics and Law, Wuhan, Hubei, China,
**2** Institution of Income Distribution and Public Finance, Zhongnan University of Economics and Law, Wuhan, Hubei, China

* yuezhang.znufe@163.com

## Abstract

### Background

The elderly in rural areas comprise over half of the older population in China, and their health problems are a matter of great concern for the Chinese government and society. Among the many factors affecting health, social capital has generated much interest in academic research. Exploring the relationship between social capital and individual health among the elderly in rural China provides ways to improve the health of Chinese people, which has a positive impact on policy.

### Methods

We selected 3719 respondents from the 2016 China Family Panel Studies (CFPS). Structural and cognitive social capital were obtained via exponentiation of variables (mean of zero and a standard deviation of one) and by giving them equal weight. Ordinary least squares (OLS) and two-stage least squares (2SLS) estimators were used to analyze the association between social capital and individual health. We explored the mechanism linking structural and cognitive social capital with individual health through a mediation effect analysis.

### Results

After correcting for endogeneity bias, structural social capital had a positive correlation with individual health among the elderly, with a coefficient of 0.062 (95% CI: 0.020-0.104). Cognitive social capital also had a positive correlation with individual health, with a coefficient of 0.097 (95% CI: 0.060-0.135). Physical exercise and positive attitude were two significant mediating variables of the relationship between social capital and individual health in the study group, with mediating effects of 0.018 and 0.054, respectively.

### Conclusions

Cognitive social capital played a stronger role than structural social capital in promoting individual health among the elderly. Physical exercise and positive attitude mediated the

**Data Availability Statement:** The data processed in this article was put into the supporting information file. The metadata of this paper comes from the China Survey Data Archive (CSDA) of

Peking University. CFPS data can be applied for and downloaded on this platform. Therefore, we did not put metadata into a public repository. If readers want to use the data, they can visit the website: https://opendata.pku.edu.cn/dataverse/CSDA.

**Funding:** This study was supported by the Planning Fund Project of Humanities and Social Sciences of Ministry of Education of China, and the funder name is "A study on the livelihood reconstruction of farmers with expropriated land in the process of urbanization: based on the framework of sustainable livelihood analysis" (grant/award number: 19YJA840006) to ZY. The funder website is http://www.moe.gov.cn/s78/A13/A13_gggs/s8474/201901/t20190129_368523.html. This study was also supported by the Surface Project of National Natural Science Foundation of China, and the funder name is "Economic Growth and Environment Pollution of Villagers under the Background of Rural Revitalization: Based on EKC Hypothesis and Evidence from Hubei" (grant/award number: 71973154) to ZY. The funder website is http://www.nsfc.gov.cn/. The two funders played roles in study design, decision to publish and preparation of manuscript.

**Competing interests:** The authors have declared that no competing interests exist.

relationship between social capital and individual health. Policymakers should not only build basic medical and health care systems but also consistently cultivate and strengthen structural and cognitive social capital among the elderly in rural China.

## Introduction

China's population is rapidly aging. By the end of 2018, the number of adults aged 65 years and over had reached 166.58 million, accounting for 11.9% of the total population [1]. With the aging of the population, the risk of disease and disability increases, and the proportion of elderly patients aged 61 and older in 2018 increased 20.5% compared with that in 2017 [2]. Older adults in rural areas constitute the majority of elderly persons in China, and a large number of them live alone at home because their family numbers go out to work in cities with the acceleration of urbanization. These old people do not have much financial income, do not get adequate medical and health services, and often take care of themselves. The main problems rural elderly face are economic poverty, poor physical health, mental loneliness and etc. With the aging of rural areas, the health problem of rural elderly is more prominent than that of urban elderly, which need significant concern of the whole society. Currently, Chinese government has carried out "Healthy China" strategy to meet the challenge of aging. Under such circumstances, knowledge of social determinants of healthy aging are crucial for the development of evidence-based policies and interventions and the sustainable development of Chinese society.

Social capital has been increasingly recognized as significant factor for health [3] and has become an important supplement to the formal medical and health services system. China is a typical Guanxi-based society, and evidence shows Guanxi and social capital have similar connotations and effects [4]. Chinese tend to seeking for social support and maintain social status in the social structure in which they live [5]. In addition, the culture of rural China values trust, mutual assistance and reciprocal exchange, which provide cultural soil for cultivating social capital. Rural residents tend to be more altruistic, honest, and trusting of others, and they reported higher levels of civic cohesion and interpersonal trust than their urban counterparts in China [6]. Some relevant departments do not provide sufficient or formal credit systems for rural elderly, and the elderly often rely more heavily on the development of social capital in daily life. This study would explore the association between social capital and individual health among the elderly in rural area. If social capital does have a promotion on individual health among the elderly in rural China, then providing them with easier access to increasing social capital will make a reference for the implementation of the "Healthy China" strategy.

### Definition and classification of social capital

As a formal concept, social capital was first proposed by French sociologist Bourdieu. He defined social capital as "a collection of actual or potential resources related to a lasting network of mutually acquiescent or recognized relationships, which are more or less institutionalized" [7]. Unlike Bourdieu, Coleman defined social capital from the perspective of function. Coleman described social capital as being imbedded in social relationships and as serving as a resource for people to achieve their goals [8]. In his opinion, except for that derived from formal organizations with a position-based structure, all forms of social capital are dependent on the stability of the social structure. The disintegration of social organization and interpersonal relationships means the breakdown of interpersonal interests. In addition, social capital loses

the premise for its existence. Putnam defined social capital as "features of social organization such as networks, norms, and social trust that facilitate coordination and cooperation for mutual benefit" [9]. Putnam actually considered social capital to be a combination of subjective social norms (trust), objective social characteristics (social networks) and outcomes (effectiveness and function). Social capital has powerful explanatory power and influence from multiple perspectives.

Regarding the classification of social capital, it has been argued that the nature of social capital is too diverse and too comprehensive to be operationalized [10]. It is necessary to deconstruct the concept of social capital into its major elements. In academic studies, it is common practice to divide social capital into structural and cognitive components. Cognitive social capital mainly includes an individual moral norms, values, attitudes and trust [11]. Structural social capital is mainly an individual social networks, his or her social engagement and other social structural factors [12]. The value of the cognitive component is generally regarded as a novel contribution of social capital theory [13]. Although the two components are related, they are distinguishable and should be separated from each other [14]. The cognitive component refers to an individual subjective perceptions of accessible social resources, whereas the structural component refers to actual social activities in formal or informal networks (civic engagement and social participation) that can be objectively measured [15]. Structural social capital is thoroughly embodied in the relationships within voluntary associations, consistent with Coleman's definition. Cognitive social capital is mainly embodied in social norms and values, especially social credit, as in Putnam's definition. In this study, we selected structural and cognitive aspects as dimensions of social capital.

## Structural, cognitive social capital and health

The importance of social capital for health has been increasingly recognized in the public health literature [16]. Scholars have found that social capital plays a significant role in self-rated health, healthy behavior and chronic illness. Regarding cognitive social capital, trust and neighborhood relationships are positively associated with self-rated health [17,18]. Structural social capital, such as social support and social networks, is associated with self-related health and better functioning [19,20], and a well-structured social network is negatively rated to individual mortality [21]. Among the elderly, social capital significantly promotes their self-rated health, activities of daily living, and chronic disease and mental health status, where its most important role is to promote activities of daily living [22]. Cognitive social capital can significantly alleviate depressive symptoms, relieve loneliness and increase life expectancy among older adults [23]. Structural social capital in the form of social networks is more likely to be associated with better mental health than with better self-related health among the elderly [24]. Social capital has been found to be associated with physical and mental health in Chinese population [25,26]. Participation in social activities promotes health among the elderly in China [27], and bonding trust and bonding networks are positively associated with older adults' physical and mental health in urban China [28]. It has already been found that structural and cognitive social capital are strongly correlated with individual health. We seek to further explore the relationship between structural and cognitive social capital and individual health among the elderly in rural areas.

## The mechanism linking social capital and individual health

Identifying the mechanisms linking social capital and health is challenging in part due to the reliance on cross-sectional data [29,30]. Several scholars have explored the mechanism linking social capital and health. Generally, social capital plays a positive role in individual health

through two means: healthy behavior and mental state. Chen H and Meng T proposed that the mechanism linking social capital and health mainly involves social support, psychological perception and material resource access [31]. Mohnen et al. found that social capital is mainly correlated with health via physical exercise and social interaction [32]. Kawachi et al. systematically summarized three possible mechanisms by which social capital affects health status: social capital affects the health of community members through the rapid dissemination of health information, increasing the frequency of exercise and other healthy behaviors; cohesive community members are more likely to create social organizations to ensure individuals' access to community health-related services, such as institutions and entertainment facilities; and social capital can improve personal health by providing emotional support and reinforcing positive social and psychological processes [33]. Scholars have focused on two key issues when analyzing the impact of social capital on health: the acquisition of resources and the adjustment of mentality. Specifically, norms of conduct from structural social capital can encourage the elderly to imitate healthy behavior and exchange health information, which is conducive to improving human physiological function and promoting the consumption of material. Cognitive social capital is often seen as emotional support. It enhances residents' awareness of disease and prevents its spread through social networks [34]. The elderly in rural areas have relationships of strong trust and dependence on the individuals around them, allowing them to participate in emotional communication and exchange of information through the social network. This is highly beneficial in helping this population manage negative emotion and adjust their physical and mental state. Thus, we mainly used healthy behavior (e.g., physical exercise) and emotional state (e.g., positive attitude) as mediators and investigated the mechanism linking social capital and individual health among the elderly in rural China.

Current studies on social capital and individual health have paid little attention to the elderly in rural China. In fact, trust and reciprocity among neighbors in rural China have an important impact on people's lives. Research on their social capital can reveal further methods to improve individual health among the elderly and make the relevant departments pay more attention to the construction of social capital in rural China. Based on the theory of social capital, this study divided social capital into structural and cognitive social capital. Current scholars typically analyze social capital from the following four perspectives: the macro-level (national, state, regional and local government); the mid-level (streets and neighborhoods); the micro-level (social networks and social participants); the individual psychological level (trust and norms). Current studies mainly focused on the mid-level and micro-level of social capital. The structural social capital discussed in this study refers primarily to participation in social networks and social organizations, which occur at the micro-level. The cognitive social capital mainly consists of trust, mutual benefit and mutual assistance, which belong to the individual psychological level. Ordinary least squares (OLS) and two-stage least squares (2SLS) estimators were used to analyze the association between social capital and individual health in this study. We explored the mechanism linking structural and cognitive social capital and individual health through a mediation effect analysis.

## Methods

### Data source

The data came from the adult questionnaire of the 2016 China Family Panel Studies (CFPS), a longitudinal survey of Chinese communities, families, and individuals. The CFPS is a nationwide and large-scale social follow-up survey implemented by the Institute of Social Science Survey, Peking University (ISSS). The CFPS officially started with a baseline survey in 2010 and then carried out three rounds of full-sample tracking surveys in 2012, 2014 and 2016. This

survey focused on the economic and noneconomic welfare of Chinese residents, including economic activities, access to education, family relations, family dynamics, population migration, physical and mental health. The target sample size of the CFPS was 16000 households, whose members were from 25 provinces/municipalities/autonomous regions of China (excluding Hong Kong, Macao, Taiwan, Xinjiang, Tibet, Qinghai, Inner Mongolia, Ningxia and Hainan) and covered 95% of the population of mainland China. The CFPS followed scientific sampling methods and guaranteed the randomness of the sample. The sampling used an implicit stratification, multi-stage, multi-level probability sampling method proportional to population size (PPS). The first two phases of CFPS sampling used official administrative divisions. The third stage used the map address method to construct the end sampling frame, and the households were selected by a circular isometric sampling method with random starting points [35]. In this study, the 2016 CFPS data were selected as the sample, and the research object was older adults in rural areas. Therefore, 3719 valid samples were obtained by screening the missing values and invalid samples for the population aged 60 and older.

## Variables

**Health evaluation index.** The main dependent variable was the "health evaluation index", which was based on self-rated health and others-rated health. Respondents assessed their health using a 5-point Likert scale (1=poor; 2=fair; 3=good; 4=very good; 5=excellent). In addition to self-rated health, this survey solicited visitors' evaluation of the health status of the respondents. Visitors reported respondents' health on a scale from 1 to 7. The higher the score was, the better the respondents' health. Based on a previous study [36], we calculated self-rated health and others-rated health (mean of zero and a standard deviation of one) and gave each of them equal weight to obtain the health evaluation index, making the health assessment more objective.

To more objectively show the difference between self-rated health and others-rated health, this study conducted a further cross-analysis of these two variables. As shown in Table 1, the first column shows five self-rated health scores, and the first row shows seven others-rated health scores. With respect to self-rated health, the number of unhealthy (scored 1) older adults (n=1169) was the highest, while the number of very healthy (scored 5) older adults (n=255) was the lowest. Unhealthy (scored 1) and general (scored 2) older adults accounted for 53.24% of the total. With respect to others-rated health, the number of older adults (n=988) with a score of 5 was the highest, and the number of older adults (n=29) with a score of 1 was the lowest. The number of older adults with a score below 4 accounted for 33.69% of the total.

There was a significant difference between self-rated health and others-rated health (Table 1), which may be because respondents may exaggerate or underestimate their own health status. Interviewers may have evaluated the health status of respondents according to how they performed in the interviews, which would have caused measurement errors. Furthermore, the hidden health risks of respondents are unique and difficult to measure. For example,

**Table 1. Cross-analysis of self-rated health and others-rated health.**

| Self-rated health / Others-rated health | 1 | 2 | 3 | 4 | 5 | 6 | 7 | Total |
|---|---|---|---|---|---|---|---|---|
| 1 | 24 | 84 | 209 | 277 | 272 | 155 | 148 | 1169 |
| 2 | 1 | 12 | 75 | 182 | 219 | 187 | 135 | 811 |
| 3 | 3 | 14 | 60 | 192 | 317 | 320 | 172 | 1078 |
| 4 | 1 | 4 | 22 | 46 | 120 | 134 | 79 | 406 |
| 5 | 0 | 2 | 10 | 35 | 60 | 85 | 63 | 255 |
| Total | 29 | 116 | 376 | 732 | 988 | 881 | 597 | 3719 |

psychological health and emotional states change often and are difficult to observe. The health evaluation index was used as the dependent variable in this study to eliminate the disadvantage that respondents reported their health too subjectively and make effective use of the information observed by interviewers. We combined and calculated self-rated health and others-rated health to make the results more scientific and reasonable.

**Structural and cognitive social capital.** Structural social capital refers to externally observable aspects of social organization, such as roles, rules, procedures and precedents [3,9], for example, civic participation or group membership [17]. In accordance with the definition above, structural social capital was measured with relational network and group membership variables in this study. Relational network was used as a proxy of a person's social network, which is an important element of structural capital. Individuals' relational network was measured by the question, "In the past 12 months, what was the total amount of money, including material goods and cash, your family spent on banquets and ceremonies?" We included the amount of money (measured in RMB yuan) in logarithm form. Organization membership was determined based on the following three questions: "Are you a member of the Communist Party of China?", "Are you a member of a religious group?" and "Are you a member of an association of individual workers?". Respondents answered "Yes" (1) or "No" (0) to each question in the interview. Cognitive social capital is more internal and subjective, referring to shared norms, values, attitudes and beliefs [3]. In accordance with the definition above, cognitive social capital consists of trust, reciprocity and mutual assistance in this study. In the 2016 CFPS, respondents were asked to indicate their trust in different people: "Let '0' be very untrustworthy and '10' be very trustworthy. Please rate your degree of trust in the following groups of people". The groups were parents, neighbors, Americans, strangers, government officials and doctors. Because Americans were not relevant to individual social capital in the study population and participants' parents may have died long ago, the trust scores for Americans and parents were excluded. The trust scores for the other groups were summed and averaged to produce a general trust score. Reciprocity was assessed by the question "How is the relationship between neighbors in your community?" Responses to the question ranged from 1 to 5. The higher the score was, the better the neighborhood relationship. Mutual assistance was measured by the question "When you need any help from neighbors, do you think anyone will give a helping hand?" Responses to the question also ranged from 1 to 5. The higher the score was, the stronger the feeling. We calculated these indicators of structural and cognitive social capital (mean of zero and a standard deviation of one) and gave them the same weight to obtain the structural social capital index and cognitive social capital index.

**Mediating variables.** We selected two mediating variables: positive attitude and physical exercise. The 2016 CFPS provided some descriptions of respondents' mental status in the past week. Positive attitude was indicated by two feelings: "I am happy in life" and "I feel pleasant." The responses ranged from 1 to 4. We added the scores and averaged them to obtain a positive attitude. Physical exercise was measured by the question "How often do you exercise a week?". Physical exercise is a numerical variable.

**Control variables.** We mainly selected demographic characteristics as control variables, including sex, age, education level, marital status and social status. Males were indicated by (1) and females by (0). Age was measured in years and centered on the mean (68.829). Education was measured by the highest level of education attained, in five categories: illiterate/semi-literate (0), primary school (1), junior high school (2), senior high school/vocational school (3), three-year college (4), 4-year college/bachelor's degree (5) and master's degree/doctoral degree (6). Marital status was assessed in two categories: never married/divorced/widowed (0) and married/cohabiting (1). Social status was measured by the question "What is your social status in your local area?" The answers ranged from low (1) to high (5).

## Statistical analysis

The characteristics of the participants were expressed as the mean and standard deviation for continuous variables and number (percentage) for categorical variables. Descriptive analysis (Table 2) showed that the structural social capital of the elderly was not high, with a mean relational network score of 6.75 and only 14.20% of older people participating in organizations. We found that the maximum (2.26) of the structural social capital index did not differ greatly from the minimum (-1.53). However, the cognitive social capital of the rural elderly was not low, as the means of reciprocity and mutual assistance were 3.90 and 4.56, respectively. It was surprising that the mean (5.38) of trust was not high, although the maximum was 10. We found that the maximum (1.20) of the cognitive social capital index was fairly distant from the minimum (-3.68). On the whole, the average of cognitive social capital (0.01) was higher than that of structural social capital (-0.01).

Regarding the statistical analysis of the other variables (Table 2), the mean positive attitude among the elderly was not low, at 2.93. However, the mean physical exercise score was only 2.40, which indicated that some older people did not usually engage in physical activities. There was a large gap between the maximum (21) and the minimum (0) for physical exercise, indicating a large difference in physical exercise involvement. Males accounted for 51.82% of the sample in this study. The average age of respondents was 68.83. It is worth noting that the mean education level among the elderly in rural areas was very low, only 0.58. Respondents with a spouse accounted for 79.65%. There was no large gap in social status among the elderly, and the mean was 3.14.

## Models

We established the following models to study the relationship between structural and cognitive social capital and individual health:

$$Y_i = \beta_1 S_i + \beta_2 C_i + \delta X_i + \varepsilon_i \qquad (1)$$

**Table 2. Descriptive statistics of the variables (N=3719).**

| Variables | Max | Min | Mean | S.D. |
|---|---|---|---|---|
| Self-rated health | 5 | 1 | 2.40 | 1.22 |
| Others-rated health | 7 | 1 | 5.03 | 1.37 |
| Health evaluation index | 1.78 | -2.03 | 0.01 | 0.79 |
| Relational network | 12.71 | 0 | 6.75 | 2.66 |
| Organization membership | Yes (14.20%); No (85.80%) | | | |
| Structural social capital index | 2.26 | -1.53 | -0.01 | 0.74 |
| Trust | 10 | 0 | 5.38 | 1.62 |
| Reciprocity | 5 | 1 | 3.90 | 0.81 |
| Mutual assistance | 5 | 1 | 4.56 | 0.89 |
| Cognitive social capital index | 1.20 | -3.68 | 0.01 | 0.70 |
| Positive attitude | 4 | 1 | 2.93 | 0.89 |
| Physical exercise | 21 | 0 | 2.40 | 3.38 |
| Sex | Males (51.82%); Females (48.18%) | | | |
| Age | 98 | 62 | 68.83 | 5.85 |
| Education level | 5 | 0 | 0.58 | 0.86 |
| Marital status | Have a spouse (79.65%); No spouse (20.35%) | | | |
| Social status | 5 | 1 | 3.14 | 1.15 |

In Model 1, $Y_i$ was the health evaluation index; $S_i$ was the structural social capital index; $C_i$ was the cognitive social capital index; $X_i$ were the control variables; and $\beta_1$ and $\beta_2$ were the estimated parameters reflecting the effect of structural and cognitive social capital on individual health, respectively. The dependent variable in this study was the health evaluation index, a numerical variable. Therefore, we used a linear regression model.

The main problem with Model 1 was the endogeneity between structural social capital and individual health. One's health condition may in turn affect one's structural social capital. It is possible that older people with better health were more inclined to expand their relational networks and participate in various organizations. Endogeneity can result in bias in the estimation results of Model 1. The best way to solve endogeneity bias is to find suitable instrumental variables. In this study, the variable of lag one phase of the endogeneity explanatory variable (structural social capital) was selected as the instrumental variable. On the one hand, the endogenous explanatory variable (structural social capital) was related to the lag variable. On the other hand, the variable of lag one phase had already occurred and may not be related to the current period. The disturbance was irrelevant. We used the relational network and organization membership of the same cohort of older adults from the 2014 CFPS data as instrumental variables.

Using two-stage least squares (2SLS) estimation, we reset the regression equation to:

$$\text{The first stage}: \ S_i = \lambda_1 Z1_i + \lambda_2 Z2_i + \gamma X_i + v_i \tag{2}$$

$$\text{The second stage}: \ Y_i = \beta_3 \bar{S}_i + \beta_4 C_i + \delta X_i + \varepsilon_i \tag{3}$$

In Model 2, $Z_1$ and $Z_2$ were instrumental variables; $X_i$ represented other control variables; $v_i$ was the linear random error; $\lambda_1$ and $\lambda_2$ were the estimated parameters, which respectively indicated the influence of the relationship network and organization membership on structural social capital. In Model 3, $\bar{S}_i$ was the predictive effect of the first-stage regression result; $C_i$ was cognitive social capital; $X_i$ represented other control variables; and $\varepsilon_i$ was the linear random error. $\beta_3$ and $\beta_4$ were estimated parameters that represented the predictive effect of the first-stage regression result and the relationship between social capital and individual health, respectively.

According to the method of causal stepwise regression proposed by Baron and Kenny [37], the procedure for testing the mediation effect is as follows: first, the independent variable is regressed on the dependent variable, and the regression coefficient c must be significant. The existence of main effect is the premise of the mediation effect (Model 4). Then, the independent variable is regressed on the intermediate variable. The regression coefficient a should be significant, which indicates that the independent variable has an effect on the intermediary variable (Model 5). Finally, the independent variable and the intermediate variable are simultaneously regressed on the dependent variable. The variable regression coefficient b should be significant (Model 6), while the independent variable regression coefficient c' is not significant or the effect size is significantly reduced compared to c. Satisfying all the above three conditions is indicative of a mediating effect. In addition, the coefficient c' in Model 6 is used to determine whether the mediation effect is partial (c' is significant) or complete (c' is not significant).

$$y_i = i + cx_i + c_1 \tag{4}$$

$$m_i = i + ax_i + c_2 \tag{5}$$

$$y_i = i + cx_i + bm_i + c_3 \tag{6}$$

In Model 4, $y_i$ was the health evaluation index; $x_i$ was structural and cognitive social capital and other control variables; and c was the estimated parameters. In Model 5, $m_i$ represented positive attitude and physical exercise; $x_i$ was structural and cognitive social capital and other control variables; and a was the estimated parameters. In Model 6, $y_i$ was the health evaluation index; $x_i$ was structural and cognitive social capital and other control variables; $m_i$ was positive attitude or physical exercise; and c and $b$ were the estimated parameters.

We estimated the models with the statistical software package Stata 13.0.

## Results

### Investigating the relationship between social capital and individual health

In Model 1 of Table 3, only the mediating variables and control variables were included to examine the relationship between them and individual health. Both mediating variables had a

**Table 3. Regression results of the relationship between social capital and individual health.**

| Variables | | Model 1 Coef. | Model 2 Coef. | Model 3 Coef. | Model 4 Coef. |
|---|---|---|---|---|---|
| Structural social capital index | | / | 0.068*** | / | 0.062** |
| | | / | (0.035-0.102) | / | (0.020-0.104) |
| Cognitive social capital index | | / | 0.097*** | -0.003 | 0.097*** |
| | | / | (0.061-0.133) | (-0.022,0.017) | (0.060-0.135) |
| Mediating variables | Positive attitude | 0.221*** | 0.203*** | 0.001 | 0.203*** |
| | | (0.193-0.248) | (0.175-0.231) | (-0.013-0.016) | (0.174-0.232) |
| | Physical exercise | 0.011** | 0.009** | 0.010*** | 0.009* |
| | | (0.004-0.018) | (0.002-0.016) | (0.006-0.014) | (0.002-0.016) |
| Control variables | Sex | 0.123*** | 0.120*** | 0.041** | 0.121*** |
| | | (0.072-0.174) | (0.069-0.171) | (0.013-0.069) | (0.070-0.171) |
| | Age | -0.011*** | -0.011*** | 0.004** | -0.011*** |
| | | (-0.016–0.007) | (-0.016–0.007) | (0.001-0.006) | (-0.016–0.007) |
| | Education level | 0.067*** | 0.059*** | 0.029*** | 0.060*** |
| | | (0.037-0.097) | (0.029-0.089) | (0.011-0.048) | (0.031-0.088) |
| | Marital status | 0.055 | 0.052** | 0.015 | 0.053 |
| | | (-0.009-0.119) | (-0.012-0.116) | (-0.018-0.047) | (-0.012-0.117) |
| | Social status | 0.067*** | 0.054*** | 0.010 | 0.055*** |
| | | (0.046-0.088) | (0.033-0.076) | (-0.002-0.022) | (0.033-0.077) |
| Instrumental variables | Relational network in lag period | / | / | 0.195*** | / |
| | | / | / | (0.190-1.999) | / |
| | Organization membership in lag period | / | / | 0.972*** | / |
| | | / | / | (0.917-1.028) | / |
| Constant | | -0.251 | -0.145 | -1.793*** | -0.146 |
| | | (-0.579-0.076) | (-0.472-0.183) | (-1.986–1.600) | (-0.472-0.180) |
| R² | | 0.117 | 0.126 | 0.715 | 0.128 |
| Number of observations | | 3719 | 3719 | 3719 | 3719 |

The 95% confidence intervals are in parentheses.

*$p<0.05$

**$p<0.01$

***$p<0.001$.

significant correlation with individual health among the elderly, with coefficients of 0.221 ($p<0.001$) and 0.011 ($p<0.001$). Age had a negative association with individual health, as the coefficient was -0.011 ($p<0.001$). Males were more likely to reporter better health than females were. Education and social status had a positive correlation with individual health; the coefficients were both 0.067 ($p<0.001$). However, marriage did not have a close relationship with individual health. In Model 2, after controlling for other factors affecting the health among the elderly, structural and cognitive social capital were added to the estimated model. Structural social capital played a significant role in promoting individual health among the elderly, with a coefficient of 0.068 ($P<0.01$). Cognitive social capital also had a significant correlation with individual health; the coefficient was 0.097 ($P<0.001$).

According to the above findings, there was an obvious correlation between structural and cognitive social capital and individual health, but any observed causal relationship must be interpreted carefully. We used 2SLS estimation, Model 3 shows the correlated factors of structural social capital. The relationship network and organizational membership in the first phase had a significant positive correlation with structural social capital. The first phase F statistic was 914.48, which was far greater than 10. There was no problem with weak instruments. The model was subjected to an over-identification test ($p=0.6647$); the relationship between networks and organization membership in the lag phase was not related to the disturbance term. In Model 4, both structural and cognitive social capital had a positive correlation with individual health, and the coefficients were 0.062 ($p<0.01$) and 0.097 ($p<0.001$), respectively. Compared with those in Model 2, the direction of each control variable did not change, and the coefficient of structural social capital on individual health dropped from 0.068 to 0.062. After correcting for endogeneity bias, the general results showed a causal relationship from structural and cognitive social capital with individual health. The estimation results also revealed that the effect of structural social capital on health was larger than that of cognitive social capital, consistent with the current academic perspective [38,39].

## Analysis of the mechanism linking social capital and individual health

The theoretical analysis and empirical results mentioned above showed relationships between structural and cognitive social capital and individual health among the elderly in rural China. To further clarify the mechanism by which social capital is correlated with individual health, we examined the mediating effects of physical exercise and positive attitude on the relationship between social capital and individual health. The results are shown in Table 4. In Model 5, structural social capital had a significant correlation with individual health ($\beta=0.085$, $P<0.001$), and cognitive social capital was significantly associated with individual health ($\beta=0.150$, $P<0.001$).

In Model 6, we examined the relationship between social capital and the mediating variable (physical exercise). Structural social capital had a positive correlation with physical exercise ($\beta=0.385$, $P<0.001$). Cognitive social capital also had a significant association with physical exercise ($\beta=0.249$, $P<0.01$). In Model 8, the independent variables and mediating variable (physical exercise) were added to the model to investigate the relationship between them and individual health among the elderly. The coefficient of structural social capital and individual health was 0.080 ($P<0.001$), and the coefficient of cognitive social capital and individual health was 0.147 ($P<0.001$). The mediating variable (physical exercise) had a significantly positive correlation with individual health ($\beta=0.012$, $P<0.01$). Physical exercise played an intermediary role in the relationship between structural social capital and individual health, and the mediating effect was 0.005 ($0.385^*0.012=0.00462$). It also played an intermediary role in the relationship between cognitive social capital and individual health, and the mediating effect was 0.003 ($0.249^*0.012=0.002988$).

**Table 4. Mediation analysis results for physical exercise and positive attitude.**

| Variables | | Health evaluation index | Physical exercise | Positive attitude | Health evaluation index | Health evaluation index |
|---|---|---|---|---|---|---|
| | | Model 5 | Model 6 | Model 7 | Model 8 | Model 9 |
| Independent variable | Structural social capital index | 0.085*** | 0.385*** | 0.064**ss | 0.080*** | 0.072*** |
| | | (0.051-0.119) | (0.238-0.532) | (0.026-0.102) | (0.046-0.114) | (0.038-0.105) |
| | Cognitive social capital index | 0.150*** | 0.249** | 0.251*** | 0.147*** | 0.099*** |
| | | (0.114-0.186) | (0.094-0.405) | (0.211-0.292) | (0.111-0.183) | (0.063-0.135) |
| Instrumental variables | Physical exercise | | | | 0.012** | |
| | | | | | (0.004-0.019) | |
| | Positive attitude | | | | | 0.205*** |
| | | | | | | (0.0177-0.233) |
| R² | | 0.078 | 0.045 | 0.074 | 0.080 | 0.126 |
| Number of samples | | 3719 | 3719 | 3719 | 3719 | 3719 |

The 95% confidence intervals are in parentheses.

*$p < 0.05$

**$p < 0.01$

***$p < 0.001$. The effect of control variables was omitted.

In Model 7, we examined the relationship between social capital and the mediating variable (positive attitude). Structural social capital had a positive correlation with positive attitude ($\beta$=0.064, $P < 0.01$). Cognitive social capital also had a significant association with positive attitude ($\beta$=0.251, $P < 0.001$). In Model 9, the independent variables and mediating variable (positive attitude) were added to the model to investigate the relationship between them and individual health among the elderly. The coefficient of structural social capital and individual health was 0.072 ($P < 0.001$). The coefficient of cognitive social capital and individual health was 0.099 ($P < 0.001$). The mediating variable (positive attitude) had a significantly positive correlation with individual health ($\beta$=0.205, $P < 0.001$). Positive attitude played an intermediary role in the relationship between structural social capital and individual health, with a mediating effect of 0.013 (0.064*0.205=0.01312). It also played an intermediary role in the relationship between cognitive social capital and individual health; the mediating effect was 0.051 (0.251*0.205=0.051455).

The above analysis results indicate that both physical exercise and positive attitude played an intermediary role in the relationship between structural and cognitive social capital and individual health. As shown in Table 5, the percentage of the mediating effect of physical exercise on the relationship between structural social capital and individual health (5.88%) was greater than the that on the relationship between cognitive social capital and individual health among the elderly (2%). The percentage of the mediating effect of positive attitude on the relationship between structural social capital and individual health (15.29%) was less than that on the relationship between cognitive social capital and individual health among the elderly

**Table 5. Effect of physical exercise and positive attitude.**

| Instrumental variables | Independent variables | Direct effect | Mediation effect | Total effect | Percentage of the mediation effect (%) |
|---|---|---|---|---|---|
| Physical exercise | Structural social capital index | 0.080 | 0.005 | 0.085 | 5.88 |
| | Cognitive social capital index | 0.147 | 0.003 | 0.150 | 2 |
| Positive attitude | Structural social capital index | 0.072 | 0.013 | 0.085 | 15.29 |
| | Cognitive social capital index | 0.099 | 0.051 | 0.150 | 34 |

(34%). In general, the mediating effect of positive attitude was stronger than that of physical exercise.

## Discussion

### Differential relationship of structural and cognitive social capital with individual health

The key findings showed that structural and cognitive social capital had a positive correlation with individual health among the elderly in rural China. This result is consistent with the large body of research emphasizing the strong link between social capital and individual health. People who have structural social capital could get available public spaces and access to mutual support, and rapid diffusion of health information and healthy norms of behavior through their clubs and associations [40]. Social groups, even those whose focus is not directly about health (e.g. religious organizations and other interest groups), tend to provide opportunities for its members to know and stay alert about health-related issues. Structural social capital could induce more collective actions, which hold promise for improving the health and well-being of the Chinese population by promoting healthy behavior [41]. As for cognitive social capital, it indicates the ability to seek for information, material, and emotional support networks, comply with social norms and peer control, trust and work closely with others in their daily activities, all of which could lead to receive adequate medical services and psychological support to buffer sufferings caused by illness. The impact of cognitive social capital is mainly psychological support through interpersonal trust and mutual assistance, which generally predicts good self-rated health [42], and similar findings were also identified in Chinese studies [43]. Therefore, cognitive social capital has different association with individual health from structural social capital. While structural aspects provide support through formal and informal institutions, cognitive social capital may increase the sense of belonging in one's community, which is beneficial for mental health [44].

Interestingly, cognitive social capital had a stronger association with individual health promotion than structural social capital in this finding. In individual health studies, cognitive social capital, i.e., trust and reciprocity, seems to have a stronger impact on health than structural aspects [45]. The elderly in rural China participated in far fewer social organizations than old adults in urban China [27,46]. The only organizations in their vicinity may be workers' associations, party organizations, etc. However, trust and mutual assistance between neighbors are common in the social life of the elderly, and their perception of neighborhood relations is stronger than that of organizational participation. The elderly in rural China have less structural social capital than cognitive social capital, so cognitive social capital has a stronger regulating effect on their health. In particular, females have difficulty participating in social organizations because of their limited education and domestic work [20]; they have less structural social capital than males, and they receive health promotion more from cognitive social capital.

Not only in rural China, but also in other rural areas, the promotion of cognitive social capital on health is greater than that of structural social capital. Cognitive social capital indicators such as trust and reciprocity were found to have larger effects on self-rated health than structural social capital indicators (e.g., social participation) [47]. Rural respondents reported higher levels of civic cohesion and trust than their urban counterparts, but they reported lower levels of community support [48]. A study in South Australia documented that rural respondents had significantly higher levels of community trust than their urban counterparts [49]. Rural elderly tend to develop cognitive social capital from community norm and trust, and they do not have a number of recreation facilities (e.g., exercise facilities, room for card games, etc) and organizations (e.g., elderly association, activity center for the elderly, etc) to get structural social capital. This finding showed

that rural resident have more cognitive social capital than structural social capital, which lead to different relationship between them with individual health.

## How structural and cognitive social capital connect with individual health

It is worth further exploring how the two types of social capital are associated with individual health. The effect of structural social capital on health promoted an interactive function in which the network may promote the exchange of health information and the sharing of resources. Larger structural social capital (e.g., social networks) can expand the source of information and promote the spread of health information among network members. Structural social capital also can be a source of self efficacy belief for finding, understanding, and using health information [50]. The elderly in rural areas who joined various organizations had common goals and norms, which may promote the cultivation and persistence of healthy behavior. This finding supports the idea that structural social capital promotes individual health, which is consistent with the academic perspective [51,52].

With respect to cognitive social capital, trust played a role primarily in reducing social conflict, promoting social harmony and improving individuals' mood. Reciprocity and mutual assistance may promote social bonds between individuals, which could reduce the cost of health information exchange for the elderly and play a significant role in promoting individual health among older adults in rural China. People who obtain relevant health information from their interpersonal networks, the media, or their government may decide to engage in health-protective action only if they trust that particular information source [53]. Cognitive social capital is significantly associated with reduced morbidity and pain perception, which is beneficial to physical health. This result is consistent with the recent finding that enhancing cognitive social capital is an effective way to alleviate pain [54].

Furthermore, our results also showed that physical activity and positive attitude play different mediation on the relationship between social capital and individual health. This is probably because structural and cognitive social capital connect with health through different ways, and they affect different aspects of health. Structural social capital mainly provides the dissemination of health information and the imitation of health behaviors for the elderly, which demonstrated the mediating effect of physical exercise on the relationship between structural social capital and individual health was greater than positive attitude. Structural social capital is mainly beneficial to physical health, and cognitive social capital play a positive role in mental health. For example, individual-level measures of social capital, including social engagement, trust, neighborhood attachment, and sense of belonging, have been shown to be negatively associated with common mental disorders [55]. Previous research found that trust and reciprocity, as components of cognitive social capital, have a stronger relationship with the mental health of older immigrants from China than does structural social capital [14]. Cognitive mainly relieve individual stress and regulate mindset to promote health, which demonstrated the mediating effect of positive attitude on the relationship between cognitive social capital and individual health was larger than physical activity. According to previous Chinese studies, trust is the main social capital element associated with health; norms of reciprocity and social networks have little influence on population health [56]. However, this study did not examine the effects of trust, reciprocity, and mutual assistance in detail, and it did not assess the degree of influence that different forms of cognitive social capital have.

## Strengths and limitations

The first strength of this study is the use of a health evaluation index based on self-rated health and others-rated health to investigate the potential relationship between two types of social

capital and individual health. Current scholars primarily focus on the physical health, mental health or self-rated health among the elderly but have not used comprehensive evaluation indicators to measure health. The second strength is the study's solving of the endogeneity problem by finding instrumental variables and using 2SLS estimation to analyze the causal relationship between social capital and individual health. The third strength is exploring the mechanism linking social capital and individual health via two important mediating variables, physical exercise and positive attitude, which was beneficial for exploring how social capital affects individual health.

There are several limitations to this study. Firstly, the measurement of health is not an easy task. At present, the common health measurement indicators for the elderly include self-rated health, activities of daily living scales, BMI index and chronic disease types. This paper obtains a health evaluation index through the exponentiation of self-rated health and others-rated health, which corrects for the subjectivity of self-rated health. However, more accurate and scientific measurements of individual health should be used in future research. The second shortcoming is that other factors affecting health are not included in the model as control variables. The factors affecting the health status of individuals go beyond living habits, economic conditions, the social environment, genetics, family status, etc. Our variables were limited according to the existing indicators of the 2016 CFPS, which meant that we could not include other variables that affect individual health. The third limitation is the handling of endogeneity bias. Although the CFPS has three periods of survey data, the social capital variables from the three phases are not exactly the same. It is impossible to use panel data. Therefore, we used the structural capital in the lag phase as an instrumental variable to overcome endogeneity bias. More research is needed to explain the dynamics of social capital and its relationship with health over multiple time periods.

## Conclusions

This study divided social capital into structural and cognitive social capital and focused on the elderly in rural China. Using the 2016 CFPS data, we drew three conclusions. Firstly, when controlling other variables affecting health, both structural and cognitive social capital had a significant positive correlation with individual health. Secondly, after resolving the endogeneity bias of structural social capital and health, structural social capital still had a significant association with health, and the net effect was smaller than before. Thirdly, physical exercise and positive attitude played mediating roles in the relationship between social capital and individual health among elderly rural residents. The mediating effect of positive attitude was greater than that of physical exercise.

This study has several policy implications. The health status of elderly persons in rural China is poor, and the relevant government entities should carry out regular physical examinations of these individuals and follow up their physical condition in a timely manner. In particular, the health department must appoint psychologists to implement interventions for this population so that the elderly can maintain a positive attitude. The medical department should assign more medical resources to villages to build a more comprehensive rural medical and health system, actively promote health knowledge among the elderly, and encourage the elderly to correct poor diet and hygiene habits.

Our findings implied that accumulating a higher stock of social capital is beneficial to health. It is necessary to take the needs of the elderly as the starting point to establish rural organizations based on local characteristics and demands and enhance enthusiasm for participation in organizations among elderly rural residents. More public resources should be devoted to supporting local activities and local organizations in order to encourage more

elderly rural residents to join neighborhood social groups. To encourage reciprocity among neighbors, policymakers should create and develop an ethos of community support by praising and rewarding those who help their neighbors. Efforts should be made to promote a sense of belonging among neighborhood residents. By strengthening community building and valuing the importance of trust and social norms, we can make cognitive social capital play a positive role in the lives of the elderly in rural areas.

Further analyses are needed to identify which particular elements of social capital may be driving the results and the effect size. The elderly in rural China are highly heterogenous, and there are different relationships between social capital and individual health in different populations of old people. For example, left-behind old adults and others have different mental health needs, and male and female elderly have differential structural social capital. The impact of social capital on the health of different groups among the elderly is worth further exploration. Future research should also explore other mediating variables in addition to physical exercise and positive attitude that contribute to the mechanism linking social capital and individual health.

## Supporting information

**S1 Data.**
(RAR)

## Acknowledgments

The authors thank the China Survey Data Archive of Peking University for the 2016 CFPS data support.

## Author Contributions

**Conceptualization:** Hang Liang, Zhang Yue.

**Formal analysis:** Erpeng Liu.

**Funding acquisition:** Zhang Yue.

**Methodology:** Hang Liang, Zhang Yue.

**Resources:** Zhang Yue, Nan Xiang.

**Software:** Hang Liang.

**Writing – original draft:** Hang Liang, Zhang Yue.

**Writing – review & editing:** Hang Liang, Zhang Yue.

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
