## [Decision Letter · Decision Letter 0]

18 Dec 2019

PONE-D-19-29520

How does social capital affect individual health among the rural elderly in China?--mediating effect analysis of physical exercise and positive attitude

PLOS ONE

Dear Dr Yue,

Thank you for submitting your manuscript to PLOS ONE. After careful consideration, we feel that it has merit but does not fully meet PLOS ONE’s publication criteria as it currently stands. Therefore, we invite you to submit a revised version of the manuscript that addresses the points raised during the review process.

1) Manuscripts submitted to **PLOS ONE** are expected to report statistical methods in sufficient detail for others to replicate the analysis performed. Ensure that results are rigorously reported in accordance with community standards and that the statistical methods employed are appropriate for the study design. Please include reliability analyses for all measures.

2) English language -  We may recommend that authors seek independent editorial help before submitting a revision. These services can be found on the web using search terms like “scientific editing service” or “manuscript editing service.”

We would appreciate receiving your revised manuscript by 17 January 2020. To enhance the reproducibility of your results, we recommend that if applicable you deposit your laboratory protocols in protocols.io, where a protocol can be assigned its own identifier (DOI) such that it can be cited independently in the future. For instructions see: http://journals.plos.org/plosone/s/submission-guidelines#loc-laboratory-protocols

We look forward to receiving your revised manuscript.

Kind regards,

Rosemary Frey

Academic Editor

PLOS ONE

Journal Requirements:

1.

2. Please clarify if the database used is anonymized. If it is not please indicate how you ensured anonymity of research subjects. Please indicate if the original database was approved by an IRB and if so, what the name of this IRB is.

3. Please clarify your financial disclosure statement to fully explain the role that the funding agencies had in the study. In particular please describe any oversight they had and how this impacted the design and execution of the study.

Reviewers' comments:

Reviewer's Responses to Questions

**Comments to the Author**

1. Is the manuscript technically sound, and do the data support the conclusions?

Reviewer #1: No

Reviewer #2: Yes

2. Has the statistical analysis been performed appropriately and rigorously? 

Reviewer #1: Yes

Reviewer #2: Yes

3. Have the authors made all data underlying the findings in their manuscript fully available?

Reviewer #1: Yes

Reviewer #2: Yes

4. Is the manuscript presented in an intelligible fashion and written in standard English?

Reviewer #1: No

Reviewer #2: No

5. Review Comments to the Author

Reviewer #1: This paper purports to examine the associations of structural and cognitive social capital with subjective health - an index based on a person's self-rated health and a rating of that person health conducted by a third person. Additionally, they assessed physical exercise and attitude as mediators. A strength of the paper is the use of 2SLS, which takes into account bidirectional relationships.

The authors have not sufficiently demonstrated an understanding of the concept of cognitive and structural social capital and their connections with the health of individuals. The authors make claims such as "scholars have had a heated discussion" but do not cite evidence of those scholars and their heated discussions. I am concerned about the validity of the structural and cognitive social capital variables. Some information on social capital this is presented in the discussion would be better suited to the introduction, and ideally, statistics should not appear in the conclusion. Policy implications for the paper are weak, for "relevant government departments should pay attention to the health status of rural elderly".

Reviewer #2: The article " How does social capital affect individual health among the rural elderly in China? --mediating effect analysis of physical exercise and positive attitude " makes an effort to advance our understanding among individual health, social capital, physical exercise and positive attitude specifically in rural elderly which shows novelty of work. The literature review is decent, the method is acceptable, English language and style are fine but minor spell check is required.

Its better to explain the theoretical approach in the last part of introduction and it should lead the reader to believe that the current study is necessary.

The methodology part should tell the reader directly whether the sample is random or not.

The research has used the scales like Health evaluation index, Structural social capital, Cognitive Social Capital, positive attitude and physical exercise, the authors are recommended to conduct reliability test on these scales for the current sample and mention reliability scores.

Authors mentioned that one question was used to measure physical exercise but its not clear which type of scale was used and what was the standard of good/poor physical exercise.

The authors analyzed the mediating effect but the model used for mediating effect is not properly described. It is advised to make it clear that which model was used.

Its better to mention study objectives again in conclusion part.

6. PLOS authors have the option to publish the peer review history of their article (what does this mean?). If published, this will include your full peer review and any attached files.

Reviewer #1: No

Reviewer #2: Yes: Azam Tariq

College of Humanities and Social Sciences

Huazhong Agricultural University, Wuhan, China

---

## [Author Response · Author response to Decision Letter 0]

13 Jan 2020

Response to Reviewers

Dear Dr./Prof. Frey,

On behalf of my co-authors, we thank you very much for giving us an opportunity to revise our manuscript, and we also appreciate the positive and constructive comments and suggestions, given by you and other reviewers, on our manuscript entitled “How does social capital affect individual health among the elderly in rural China?--mediating effect analysis of physical exercise and positive attitude” (ID:PONE-D-19-29520).

Having studied the reviewer’s comments carefully, we have tried out best to make revisions which were marked in red in this paper. Please find the revised version enclosed, which we would like to submit for your kind consideration.

Point 1:Manuscripts submitted to PLOS ONE are expected to report statistical methods in sufficient detail for others to replicate the analysis performed. Ensure that results are rigorously reported in accordance with community standards and that the statistical methods employed are appropriate for the study design. Please include reliability analyses for all measures.

Response 1: 

We appreciate the comments of the reviewers. In the previous manuscript, we did not describe the mediation effect analysis method in detail. In the revised manuscript, we explained this method in detail (please see line 338-361). In the previous model analysis, we reported standard errors. After studying the community standards in detail, we reported 95% confidence intervals (please see Table 3 and Table 4). Regarding the data of this article, we have uploaded the data in the previous manuscript, and filled in the website link of downloading and viewing database. We added the command data in a file uploaded to ensure that the experiment can be repeated operated. We proofread the format in this study again, responded to the comments of reviewers, and revised in accordance with their suggestion. (please see below response for comments to reviewers).

Point 2:English language --We may recommend that authors seek independent editorial help before submitting a revision. These services can be found on the web using search terms like “scientific editing service” or “manuscript editing service.”

Response 2: 

Thanks a lot for the editor's suggestion. We apologize for not using the standard English language before. We have seek independent editorial help before submitting a revision.

We have invited experts whose native language is English to modify the presentation of this article, considering that these modifications may not fully meet the requirements of PLOS ONE. At the same time, we are afraid that the paper editing service organization cannot complete the paper editing within the deadline. Therefore, we promise that, in addition to the English expression, if the entire article can be officially accepted, we will use your company’s English editing service to make the manuscript fully meet the publishing requirements.

We would like to express our great gratitude to you and other reviewers for the comments on our paper. Looking forward to hearing from you.

Thank you and best regards.

Yours sincerely,

Zhang Yue

Corresponding author: 

Name: Zhang Yue

E-mail: yuezhang.znufe@163.com

Response to Reviewer 1 Comments

Dear Editors and Reviewers,

Thank you for your letter and for the reviewers’ comments concerning our manuscript entitled “How does social capital affect individual health among the elderly in rural China?--mediating effect analysis of physical exercise and positive attitude” (ID: PONE-D-19-29520). Those comments are all valuable and very helpful for revising and improving our paper, and simultaneously they are of great significance to give instructions to our research. We have studied the comments carefully and made revisions which we hope to meet with approval. Revised portions are marked in red in the original paper. The main revisions on this paper, along with our responses to the reviewer’s comments, are as follows:

Point 1: The authors have not sufficiently demonstrated an understanding of the concept of cognitive and structural social capital and their connections with the health of individuals.

Response 1: 

We appreciate the comments of the reviewers. We apologize for ignoring detailed explain the concept of cognitive and structural. In our study, cognitive social capital mainly includes an individual's moral norms, value, attitude and trust. Structural social capital is mainly an individual's social networks, his or her social engagement and other social structural factors. In this article, we further point out the characteristics and differences between structured social capital and cognitive social capital (see line 89-95). As for the concept of the two types of social capital, we mainly borrowed academic points of these references as follows: 

1.Park M. Impact of social capital on depression trajectories of older women in Korea. Aging & Mental Health, 2017,21(4):354-361. https://doi.org/10.1080/13607863.2015.1088511 PMID: 26404493

Structural social capital acts as network resources to older women such as the quantity of social network, family and friends support. The cognitive component of social capital includes perception of support, reciprocity, sharing and trust.

2.Harpham T, Grant E, Thomas E. Measuring social capital within health surveys: key issues. Health Policy Plan. 2002; 17(1):106-11. https://doi.org/10.1093/heapol/17.1.106 PMID: 11861592

 The structural component includes extent and intensity of association links or activity ,and the cognitive component covers perceptions of support, reciprocity, sharing and trust. 

3.Rostila, M. Social capital and health in European welfare regimes: a multilevel approach. Journal of European Social Policy. 2007, 17, 223-239. https://doi.org/10.1177/0958928707078366

The structural component of social capital is the extent and intensity of participation in associations and other forms of social activities (e.g. density of civic associations, measures of informal social participation) whereas the cognitive component is about peoples perceptions of interpersonal trust, sharing and reciprocity.

4.Harpham T. The measurement of community social capital trough Surveys. In: Kawachi I, Subramanian SV, Kim D, editors. Social Capital and Health. New York: Springer; 2008. p51-62.

Structural social capital refers to what people do (associational links, networks) which could be objectively verified (by observation or records). Cognitive social capital refers to what people feel (values and perceptions) and is thus subjective.

Point 2: The authors have not sufficiently demonstrated cognitive and structural social capital and their connections with the health of individuals.

Response 2: 

In the previous manuscripts, we mainly focused on the literature review of the relationship between social capital and health and did not review the literature of structured social capital, cognitive social capital’s connection with health in more detail. We reorganized the relevant literature and focused on solving this problem. Please see line105-118.

List of newly added related references as follows:

1.Ichida Y, Kondo K, Hirai H, et al. Social capital, income inequality and self-rated health in Chita Peninsula, Japan: a multilevel analysis of older people in 25 communities. Soc Sci Med 2009;69:489-99 https://doi.org/10.1016/j.socscimed.2009.05.006 PMID: 19523728

People who live in conditions of high-income inequality tend to exhibit low trust levels, and that trust of cognitive social capital mediates the relation between income inequality and health.

2.Pollack CE and von dem Knesebeck O. Social capital and health among the aged: Comparisons between the United States and Germany. Health Place. 2004; 10(4):383-91. https://doi.org/10.1016/j.healthplace.2004.08.008 PMID: 15491897 

social capital including both norms (reciprocity and civic trust) and behaviors (participation) were associated with self-reported health (indicators-overall health, depression (CES-D) and functional limitations). 

3.Li Q, Zhou X, Ma S and etc. The effect of migration on social capital and depression among older adults in China. Soc Psychiatry Psychiatr Epidemiol. 2017 Dec;52(12):1513-1522. https://doi.org/10.1007/s00127-017-1439-0. 

Social capital measurements included cognitive (generalized trust and reciprocity) and structure (support from individual and social contact) aspects.The depression disadvantage is partly accounted for by lower level of cognitive social capital (trust and reciprocity).

4.Brinkhues S, Dukers-Muijrers NHTM, Hoebe CJPA, et al. Socially isolated individuals are more prone to have newly diagnosed and prevalent type 2 diabetes mellitus-the Maastricht study-. BMC Public Health. 2017; 17:955. https://doi.org/10.1186/s12889-017-4948-6 PMID: 29254485

Several aspects of structural and functional characteristics of the social network were associated with newly and previously diagnosed T2DM, Men and women who were more socially isolated, and who received less emotional and practical support, more frequently had newly and previously diagnosed T2DM.

5.Liu G, Xue X, et al. How does social capital matter to the health status of older adults? Evidence from the China Health and Retirement Longitudinal Survey. Econ Hum Biol. 2016; 22:177-189. https://doi.org/10.1016/j.ehb.2016.04.003 PMID: 27235837

We obtain evidence indicating that structural social capital has a significant and positive effect on general and physical health but not on mental health.

6.Norstrand JA, Xu Q. Social capital and health outcomes among older adults in China: the urban-rural dimension. Gerontologist. 2012; 52(3):325-34. https://doi.org/10.1093/geront/gnr072 PMID: 21746837 

Trust was significantly associated with physical and emotional health, and participation of organization was significantly associated with physical health among the urban older Chinese.

Point 3: The authors make claims such as "scholars have had a heated discussion" but do not cite evidence of those scholars and their heated discussions.

Response 3: 

We appreciate the comments of the reviewers. We apologize for exaggerating current research on the mechanism that correlates social capital with individual health. We have corrected the previous statement. Please see line 124-125. 

We have combed through related literature and found that there are not enough articles on the mechanism of how social capital affects health. We found four articles which propose the mechanism that connect social capital with individual health. Please see references 29,30,31,32. We think that instead of finding their argument here, we found some common points of the predecessors, which can contribute to the mechanism in this study. Please see line 138-139.

Point 4: I am concerned about the validity of the structural and cognitive social capital variables.

Response 4: 

We appreciate the comments of the reviewers. In the part of variable selection, we did not introduce the basis of selecting variables of structural and cognitive social capital in detail, which led to the lack of effectiveness in selecting related variables. Structural social capital refers to externally observable aspects of social organization, such as roles, rules, procedures and precedents. Cognitive social capital is more internal and subjective, referring to shared norms, values, attitudes and beliefs. We have added the selection basis of these two types of social capital variables in the variables section to ensure the validity of variable selection. Please see line 229-231, 242-243. The references are listed as follows:

1.Moore S, Kawachi I. Twenty years of social capital and health research: a glossary. J Epidemiol Community Health. 2017; 71(5):513-7. https://doi.org/10.1136/jech-2016-208313 PMID: 28087811

2.Putnam RD. Bowling Alone: America's Declining Social Capital. Journal of Democracy. 1995; (January): 65-78.

3.Dauner KN, Wilmot NA, Schultz JF. Investigating the temporal relationship between individual-level social capital and health in fragile families. BMC Public Health. 2015; (15): 1130. https://doi.org/10.1186/s12889-015-2437-3 PMID: 26572491

Point 5: Some information on social capital this is presented in the discussion would be better suited to the introduction.

Response 5: 

We agree with the reviewer’s suggestion that some information on social capital presented in the discussion would be better suited to the introduction. We found that analysis perspectives of social capital are more suitable for the introduction. Please see line 158-166.

Point 6: Statistics should not appear in the conclusion.

Response 6: 

We appreciate the comments of the reviewers. We apologize for Putting the statistics in the conclusion part of this article. We have deleted the statistics in the conclusion section, please see line 544-548.

Point 7: Policy implications for the paper are weak, for "relevant government departments should pay attention to the health status of rural elderly".

Response 7:

We agree with the comments of the reviewers. We have rewritten the policy recommendations section. It is mainly carried out from two aspects. One is that how the governments effectively manage and timely detect individual health among the elderly in rural China. The other is how the relevant departments effectively promote the increase of rural elderly's structural and cognitive social capital. In this regard, we make the following suggestions:

The health status of elderly persons in rural China is poor, and the relevant government entities should carry out regular physical examinations of these individuals and follow up their physical condition in a timely manner. In particular, the health department must appoint psychologists to implement interventions for this population so that the elderly can maintain a positive attitude. The medical department should assign more medical resources to villages to build a more comprehensive rural medical and health system, actively promote health knowledge among the elderly, and encourage the elderly to correct poor diet and hygiene habits.

More public resources should be devoted to supporting local activities and local organizations in order to encourage more elderly rural residents to join neighborhood social groups. To encourage reciprocity among neighbors, policymakers should create and develop an ethos of community support by praising and rewarding those who help their neighbors. Efforts should be made to promote a sense of belonging among neighborhood residents. By strengthening community building and valuing the importance of trust and social norms, we can make cognitive social capital play a positive role in the lives of the elderly in rural areas.

For the part we modified, please see line 552-560, 573-582.

Response to Reviewer 2 Comments

Dear Editors and Reviewers,

Thank you for your letter and for the reviewers’ comments concerning our manuscript entitled “How does social capital affect individual health among the elderly in rural China?--mediating effect analysis of physical exercise and positive attitude” (ID: PONE-D-19-29520). Those comments are all valuable and very helpful for revising and improving our paper, and simultaneously they are of great significance to give instructions to our research. We have studied the comments carefully and made revisions which we hope to meet with approval. Revised portions are marked in red in the original paper. The main revisions on this paper, along with our responses to the reviewer’s comments, are as follows:

Point 1: Its better to explain the theoretical approach in the last part of introduction and it should lead the reader to believe that the current study is necessary.

Response 1: 

We agree with the comments of the reviewers. In the previous manuscript, we failed to fully explain the theoretical method and necessity of the research in the last paragraph of introduction. In this regard, we have seriously revised and added the research significance, the analysis perspective of social capital, and the research method of this article, as detailed below:

Current studies on social capital and individual health have paid little attention to the elderly in rural China. In fact, trust and reciprocity among neighbors in rural China have an important impact on people's lives. Research on their social capital can reveal further methods to improve individual health among the elderly and make the relevant departments pay more attention to the construction of social capital in rural China. Based on the theory of social capital, this study divided social capital into structural and cognitive social capital. Current scholars typically analyze social capital from the following four perspectives: the macro-level (national, state, regional and local government); the mid-level (streets and neighborhoods); the micro-level (social networks and social participants); the individual psychological level (trust and norms). Current studies are mainly focused on the mid-level and micro-level of social capital. The structural social capital discussed in this study refers primarily to participation in social networks and social organizations, which occurs at the micro-level. The cognitive social capital mainly consists of trust, mutual benefit and mutual assistance, which belong to the individual psychological level. Ordinary least squares (OLS) and two-stage least squares (2SLS) estimators were used to analyze the association between social capital and individual health in this study. We explored the mechanism linking structural and cognitive social capital and individual health through a mediation effect analysis.

As for specific modifications, please see line 152-170.

Point 2: The methodology part should tell the reader directly whether the sample is random or not.

Response 2: 

We strongly agree with the reviewer’s recommendations. We apologize for ignoring introducing the selection process of the sample in the part of data source. 

CFPS followed scientific sampling methods and guaranteed the randomness of the sample. CFPS sampling used Implicit stratification, multi-stage, multi-level, probability sampling method proportional to population size (PPS). The first two phases of CFPS sampling used official administrative divisions. The third stage used maps address method to construct the end sampling frame, and the households were selected by using a circular isometric sampling method with random starting points. More detailes about sampling of CFPS see http://www.isss.pku.edu.cn/cfps/

In PLoS ONE, there is also study using CFPS data. (Chen H, Meng T. Bonding, Bridging, and Linking Social Capital and Self-Rated Health among Chinese Adults: Use of the Anchoring Vignettes Technique. PLoS One. 2015; 10(11):e0142300. https://doi.org/10.1371/journal.pone.0142300 PMID: 26569107）

In the revised draft, we clearly stated whether the sample selection was random. Please see line 186-191.

Point 3: The research has used the scales like Health evaluation index, Structural social capital, Cognitive Social Capital, positive attitude and physical exercise, the authors are recommended to conduct reliability test on these scales for the current sample and mention reliability scores.

Response 3: 

Thanks to the reviewer’s comments. We apologize for not providing for reliability scores of these five variables. The selection of these five indicators is based on relevant definitions rather than scales. These indicators were obtained via exponentiation of variables (mean of zero and a standard deviation of one) and by giving them equal weight.Regarding the indexing process, we are mainly for the convenience of the intermediary analysis later. For specific methods, we have also referred to relevant literature (please see reference 34. Ho CY, Better Health With More Friends: The Role of Social Capital in Producing Health. Health Econ. 2016; 25(1):91-100. https://doi.org/10.1002/hec.3131 PMID: 25431183).

The health evaluation index is obtained by self-rated health and others-rated health through operationalization and giving them equal weight. Both self-rated health and others-rated health are measures of individual health.

The structural social capital is obtained by relationship network and organization members through operationalization and giving them equal weight. The organization members and the relationship network are selected according to the traditional definition of structural social capital (please see line 229-231). 

The cognitive social capital is obtained by trust, reciprocity, and mutual assistance through operationalization and giving them equal weight. These three variables are also selected according to the traditional definition of cognitive social capital (please see line 242-243). 

Positive mentality is summed up through the equal empowerment of two questions, "I am happy in life" and "I am pleasant". One of these two questions asks the mentality in life, the other is the current subjective feeling, which measures the positive attitude in different aspects.

Physical exercise is judged by "How often do you exercise a week". There is one indicator of physical exercise.

Because the measurement of these variables is not from a scale, but specific problems. These questions are for different aspects of the indicators. We are sorry that we could not get reliability scores.

Point 4: Authors mentioned that one question was used to measure physical exercise but its not clear which type of scale was used and what was the standard of good/poor physical exercise.

Response 4: 

“Physical exercise”is judged by "How often do you exercise a week". This question is asking how many times the elderly exercise a week in rural China. But the variable does not measure good or bad of the elderly’s exercise. It is difficult to judge the standard of physical activity among the elderly in rural China. The elderly may exercise a lot, but they are doing farm work and may not have good health. So this question can only objectively reflect the number of times that the elderly exercise and can not provide measure standard. As for how it relates to health, it needs to be objectively put into the model for further investigation.

Point 5: The authors analyzed the mediating effect but the model used for mediating effect is not properly described. It is advised to make it clear that which model was used.

Response 5: 

We agree with the reviewer’s suggestion and we added analysis method of mediation effect in the part of model introduction. This method mainly refers to the practice of Baron and Kenny (Baron RM, Kenny DA. The moderator-mediator variable distinction in social psychological research: conceptual, strategic, and statistical considerations. J Pers Soc Psychol. 1986；51(6):1173-82. https://doi.org/10.1037//0022-3514.51.6.1173 PMID: 3806354). We performed stepwise regression on related variables, observes the change of coefficients, and judges whether there is a mediation effect and the size according to the significance of the variable and the size of the coefficient. The main method is shown in line 338-361.

Point 6: Its better to mention study objectives again in conclusion part.

Response 6: 

We strongly agree with this comment from the reviewers. In the conclusion part of this study, we have proposed study objectives, mainly to further subdivide Chinese rural elderly, such as examining the different impact of social capital on the health of left-behind and non-left-behind elderly in rural China, etc. And we will continue to explore the mediating variables that social capital affects the health of the elderly. Explained as follows:

Further analyses are needed to identify which particular elements of social capital may be driving the results and the effect size. The elderly in rural China are highly heterogenous, and there are different relationships between social capital and individual health in different populations of old people. For example, left-behind old adults and others have different mental health needs, and male and female elderly persons have differential structural social capital. The impact of social capital on the health of different groups among the elderly is worth further exploration. Future research should also explore other mediating variables in addition to physical exercise and positive attitude that contribute to the mechanism linking social capital and individual health.

The specific content of the modification, please see line 573-582.

---

## [Decision Letter · Decision Letter 1]

25 Feb 2020

PONE-D-19-29520R1

How does social capital affect individual health among the elderly in rural China?--mediating effect analysis of physical exercise and positive attitude

PLOS ONE

Dear Zhang,

Thank you for submitting your manuscript to PLOS ONE. After careful consideration, we feel that it has merit but does not fully meet PLOS ONE’s publication criteria as it currently stands. Therefore, we invite you to submit a revised version of the manuscript that addresses the points raised during the review process.

Please address the minor issues raised by reviewer 3 regarding:

the relationship between social capital and health outcomes

greater detail about the population under study

We would appreciate receiving your revised manuscript by 24 March 2020. To enhance the reproducibility of your results, we recommend that if applicable you deposit your laboratory protocols in protocols.io, where a protocol can be assigned its own identifier (DOI) such that it can be cited independently in the future. For instructions see: http://journals.plos.org/plosone/s/submission-guidelines#loc-laboratory-protocols

We look forward to receiving your revised manuscript.

Kind regards,

Rosemary Frey

Academic Editor

PLOS ONE

Reviewers' comments:

Reviewer's Responses to Questions

**Comments to the Author**

1. If the authors have adequately addressed your comments raised in a previous round of review and you feel that this manuscript is now acceptable for publication, you may indicate that here to bypass the “Comments to the Author” section, enter your conflict of interest statement in the “Confidential to Editor” section, and submit your "Accept" recommendation.

Reviewer #2: All comments have been addressed

Reviewer #3: All comments have been addressed

2. Is the manuscript technically sound, and do the data support the conclusions?

Reviewer #2: Yes

Reviewer #3: Partly

3. Has the statistical analysis been performed appropriately and rigorously? 

Reviewer #2: Yes

Reviewer #3: I Don't Know

4. Have the authors made all data underlying the findings in their manuscript fully available?

Reviewer #2: Yes

Reviewer #3: Yes

5. Is the manuscript presented in an intelligible fashion and written in standard English?

Reviewer #2: Yes

Reviewer #3: Yes

6. Review Comments to the Author

Reviewer #2: the authors have addressed all the questions and suggestions. and it is suggested to accept the manuscript.

Reviewer #3: This submission is a review from a original previous submission on social capital and depressive syntoms in Chinese elderly which I had not reviewed.

I have carefully read the R1 version of the paper as well as the comments by the previous reviewers and the authors answers to them. The vast majority of their comments have been addressed and my only suggestion would be to be more specific in the section describing the evidence on the relationship between social capital in its structural and cognitive dimensions and different health outcomes. This part would benefit greatly if the authors could specify the population in which the studies had been conducted - epsecially taking into account that they argue that it has been little explored in elderly popultion from China.

7. PLOS authors have the option to publish the peer review history of their article (what does this mean?). If published, this will include your full peer review and any attached files.

Reviewer #2: Yes: Azam Tariq, Department of Sociology, College of Humanities and Social Sciences, Huazhong Agricultural University,

Wuhan 430070, China; azam_tariq@webmail.hzau.edu.cn

Reviewer #3: No

---

## [Author Response · Author response to Decision Letter 1]

13 Mar 2020

Response to Reviewers

Dear Dr./Prof. Frey,

On behalf of my co-authors, we thank you very much for giving us an opportunity to revise our manuscript, and we also appreciate the positive and constructive comments and suggestions, given by you and other reviewers, on our manuscript entitled “How does social capital affect individual health among the elderly in rural China?--mediating effect analysis of physical exercise and positive attitude” (ID:PONE-D-19-29520).

Having studied the reviewer’s comments carefully, we have considered your suggestions and comments, trying our best to make revisions which were marked in red in this paper. Please find the revised version enclosed, which we would like to submit for your kind consideration.

Point 1:Please address the minor issues raised by reviewer 3 regarding:the relationship between social capital and health outcomes.

Response 1: 

We appreciate the comments of the reviewers. In our previous manuscript, we only discussed the relationship between structural and cognitive social capital during the literature review and data analysis process. In the discussion section of this revised manuscript, based on the results of empirical analysis, we conducted a more detailed discussion on the relationship between structural and cognitive social capital and health. 

Firstly, we find that both structural social capital and cognitive social capital have a significant promotion effect on health, and the coefficients were 0.062 (p<0.01) and 0.097 (p<0.001), respectively. We explain the role of structural and cognitive social capital in promoting health (please see line 475-493). 

Secondly, we find that the structural social capital's promotion of individual health among rural elderly is less than the cognitive social capital. We have explored this phenomenon in more depth. It may be because rural elderly rely more on mutual help and interpersonal trust in the community. They have more cognitive social capital than structural social capital, which has led to the different promotion of two types of social capital on individual health (please see line 496-509). We also find this difference exist in research from other regions. We analyzed our finding and results of other scholars, summarizing the reason. Rural elderly tend to develop cognitive social capital from community norm and trust, and they do not have a number of recreation facilities (e.g., exercise facilities, room for card games, etc) and organizations (e.g., elderly association, activity center for the elderly, etc) to get structural social capital (please see line 510-523).

Finally, we explore how two aspects of social capital connect with individual health (please see line 526-547). We found positive mentality and physical activity both play a mediating role on the relationship between social capital and health based on previous empirical research, but their mechanisms are different. This is probably because structural and cognitive social capital connect with health through different ways, and they affect different aspects of health. Structural social capital mainly promotes health through team members' access to health information and imitating health behaviors. Cognitive mainly relieve individual stress and regulate mindset to promote health (please see line 548-570)

We hope to discuss the above three parts to further analyze the relationship between structural social capital and individual health.

Point 2:Please address the minor issues raised by reviewer 3 regarding: greater detail about the population under study

Response 2: 

Thanks a lot for the editor's suggestion. We apologize for not give detail introduction about the elderly in rural China. Older adults in rural areas constitute the majority of elderly persons in China, and a large number of them do not have much financial income, do not get adequate medical and health services, and often take care of themselves. The main problems rural elderly face are economic poverty, poor physical health, mental loneliness and etc. With the aging of rural areas, the health problem of rural elderly is more prominent than that of urban elderly, which need significant concern of the whole society (please see line 54-65).

The living environment of the elderly in rural China is special. The culture of rural China values trust, mutual assistance and reciprocal exchange, which provide cultural soil for cultivating social capital. Rural residents tend to be more altruistic, honest, and trusting of others, and they reported higher levels of civic cohesion and interpersonal trust than their urban counterparts in China. Under such circumstances, it is necessary and important to study social capital and individual health among the elderly.

We would like to express our great gratitude to you and other reviewers for the comments on our paper. Looking forward to hearing from you.

Thank you and best regards.

Yours sincerely,

Zhang Yue

Corresponding author: 

Name: Zhang Yue

E-mail: yuezhang.znufe@163.com

Response to Reviewer 3 Comments

Dear Reviewers,

Thank you for your letter and for the reviewers’ comments concerning our manuscript entitled “How does social capital affect individual health among the elderly in rural China?--mediating effect analysis of physical exercise and positive attitude” (ID: PONE-D-19-29520). Those comments are all valuable and very helpful for revising and improving our paper, and simultaneously they are of great significance to give instructions to our research. We have studied the comments carefully and made revisions which we hope to meet with approval. Revised portions are marked in red in the original paper. The main revisions on this paper, along with our responses to the reviewer’s comments, are as follows:

Point 1:More specific in the section describing the evidence on the relationship between social capital in its structural and cognitive dimensions and different health outcomes.

Response 1: 

We appreciate the comments of the reviewer. In previous manuscripts, we only introduced the relationship between social capital and health in the literature review and empirical research sections. We apologize for ignoring detailed explaining the relationship between social capital and individual health. We have improved the manuscript and focused on the discussion to further explain the relationship between social capital and individual health. Our analysis follows the following lines: Is there a relationship between structural social capital and individual health? What are the coefficients of structural and cognitive social capital on individual health? How do structural and cognitive social capital connect with individual health?

Firstly, we find that both structural social capital and cognitive social capital have a significant promotion effect on health, and the coefficients were 0.062 (p<0.01) and 0.097 (p<0.001), respectively. We explain the promotion of structural social capital. People who have structural social capital could get available public spaces and access to mutual support, and rapid diffusion of health information and healthy norms of behavior through their clubs and associations (please see line 475-487). We also explain the role of cognitive social capital in promoting health. The impact of cognitive social capital is mainly psychological support through interpersonal trust and mutual assistance, which generally predicts good self-rated health (please see line 485-493).

Secondly, we explained why the coefficient of cognitive social capital on individual health is larger than structural social capital. The elderly in rural China participated in far fewer social organizations than old adults in urban China. Trust and mutual assistance between neighbors are common in the social life of rural elderly, and their perception of neighborhood relations is stronger than that of organizational participation (please see line 496-509). We also find this difference exist in research from other regions and summarized the reason: rural elderly tend to develop cognitive social capital from community norm and trust, and they do not have a number of recreation facilities (e.g., exercise facilities, room for card games, etc) and organizations (e.g., elderly association, activity center for the elderly, etc) to get structural social capital (please see line 510-523).

Finally, we explored that how structural and cognitive social capital connect with individual health. We focused on two different ways in which social capital affects health. Structural social capital mainly provides the dissemination of health information and the imitation of health behaviors for the elderly, which demonstrated the mediating effect of physical exercise on the relationship between structural social capital and individual health was greater than positive attitude. Structural social capital is mainly beneficial to physical health, and cognitive social capital play a positive role in mental health. Cognitive mainly relieve individual stress and regulate mindset to promote health, which demonstrated the mediating effect of positive attitude on the relationship between cognitive social capital and individual health was larger than physical activity (please see line 548-570).

We hope to explain the different relationship between structural and cognitive social capital and individual health through discussion of these three aspects above.

Point 2:This part would benefit greatly if the authors could specify the population in which the studies had been conducted - epsecially taking into account that they argue that it has been little explored in elderly popultion from China.

Response 2: 

We agree with the reviewer’s suggestion. In previous manuscripts, we did not give a detailed introduction to the group of elderly people in rural China. In fact, the health problems of this group are worthy of attention (please see line 53-65). We introduce this group as follows:

Older adults in rural areas constitute the majority of elderly persons in China, and a large number of them live alone at home because their family numbers go out to work in cities with the acceleration of urbanization. These old people do not have much financial income, do not get adequate medical and health services, and often take care of themselves. The main problems rural elderly face are economic poverty, poor physical health, mental loneliness and etc. With the aging of rural areas, the health problem of rural elderly is more prominent than that of urban elderly, which need significant concern of the whole society. Currently, Chinese government has carried out “Healthy China” strategy to meet the challenge of aging. Under such circumstances, knowledge of social determinants of healthy aging are crucial for the development of evidence-based policies and interventions and the sustainable development of Chinese society.

The environment in which they live is rich in social capital, and social capital is closely related to the lives of these elderly people. We explain the importance of studying the health and social capital issues of this group as follows :

China is a typical Guanxi-based society, and evidence shows Guanxi and social capital have similar connotations and effects. Chinese tend to seeking for social support and maintain social status in the social structure in which they live. In addition, the culture of rural China values trust, mutual assistance and reciprocal exchange, which provide cultural soil for cultivating social capital. Rural residents tend to be more altruistic, honest, and trusting of others, and they reported higher levels of civic cohesion and interpersonal trust than their urban counterparts in China. Some relevant departments do not provide sufficient or formal credit systems for rural elderly, and the elderly often rely more heavily on the development of social capital in daily life.

For more details, please see line 68-81.

We hope to explore the association between social capital and individual health among the elderly in rural area and make a reference for the implementation of the “Healthy China” strategy.

Thanks again for the reviewers' comments and suggestions!

---

## [Decision Letter · Decision Letter 2]

23 Mar 2020

How does social capital affect individual health among the elderly in rural China?--mediating effect analysis of physical exercise and positive attitude

PONE-D-19-29520R2

Dear Dr. Yue,

We are pleased to inform you that your manuscript has been judged scientifically suitable for publication and will be formally accepted for publication once it complies with all outstanding technical requirements.

With kind regards,

Rosemary Frey

Academic Editor

PLOS ONE

Additional Editor Comments (optional):

Reviewers' comments:

Reviewer's Responses to Questions

**Comments to the Author**

1. If the authors have adequately addressed your comments raised in a previous round of review and you feel that this manuscript is now acceptable for publication, you may indicate that here to bypass the “Comments to the Author” section, enter your conflict of interest statement in the “Confidential to Editor” section, and submit your "Accept" recommendation.

Reviewer #3: All comments have been addressed

2. Is the manuscript technically sound, and do the data support the conclusions?

Reviewer #3: (No Response)

3. Has the statistical analysis been performed appropriately and rigorously? 

Reviewer #3: (No Response)

4. Have the authors made all data underlying the findings in their manuscript fully available?

Reviewer #3: (No Response)

5. Is the manuscript presented in an intelligible fashion and written in standard English?

Reviewer #3: (No Response)

6. Review Comments to the Author

Reviewer #3: I commend the authors for their throughness in addressing my comments and suggestions, which have been all considered. I would only recommend a brief explanation of what a "guanxi" society means, as not all readers may be familiar with this term.

7. PLOS authors have the option to publish the peer review history of their article (what does this mean?). If published, this will include your full peer review and any attached files.

Reviewer #3: No

---

## [Editor Report · Acceptance letter]

14 Jul 2020

PONE-D-19-29520R2 

How does social capital affect individual health among the elderly in rural China?--mediating effect analysis of physical exercise and positive attitude 

Dear Dr. Yue:

I'm pleased to inform you that your manuscript has been deemed suitable for publication in PLOS ONE. Congratulations! Your manuscript is now with our production department. 

Kind regards, 

on behalf of

Dr. Rosemary Frey 

Academic Editor

PLOS ONE